# The mitochondria-targeted derivative of the classical uncoupler of oxidative phosphorylation carbonyl cyanide *m*-chlorophenylhydrazone is an effective mitochondrial recoupler

**Iliuza R. Iaubasarova[1,2], Ljudmila S. Khailova[1], Alexander M. Firsov[1], Vera G. Grivennikova[3], Roman S. Kirsanov[1], Galina A. Korshunova[1], Elena A. Kotova[1], Yuri N. Antonenko[1] ***

**1** Belozersky Institute of Physico-Chemical Biology, Lomonosov Moscow State University, Moscow, Russia, **2** Faculty of Chemistry, Lomonosov Moscow State University, Moscow, Russia, **3** Faculty of Biology, Lomonosov Moscow State University, Moscow, Russia

* antonen@belozersky.msu.ru

## Abstract

The synthesis of a mitochondria-targeted derivative of the classical mitochondrial uncoupler carbonyl cyanide-m-chlorophenylhydrazone (CCCP) by alkoxy substitution of CCCP with n-decyl(triphenyl)phosphonium cation yielded mitoCCCP, which was able to inhibit the uncoupling action of CCCP, tyrphostin A9 and niclosamide on rat liver mitochondria, but not that of 2,4-dinitrophenol, at a concentration of 1–2 µM. MitoCCCP did not uncouple mitochondria by itself at these concentrations, although it exhibited uncoupling action at tens of micromolar concentrations. Thus, mitoCCCP appeared to be a more effective mitochondrial recoupler than 6-ketocholestanol. Both mitoCCCP and 6-ketocholestanol did not inhibit the protonophoric activity of CCCP in artificial bilayer lipid membranes, which might compromise the simple proton-shuttling mechanism of the uncoupling activity on mitochondria.

## 1. Introduction

It is now generally accepted that electron transfer via a chain of proton pumps in the inner mitochondrial or bacterial membrane results in the formation of a transmembrane difference of electrochemical potentials of hydrogen ions that couples the oxidation of respiratory substrates to ATP synthesis. Compounds that compromise the coupling between respiration and phosphorylation called uncouplers have recently received strong interest as promising anti-cancer, antibacterial, anti-obesity, antidiabetic, neuroprotective and cardioprotective agents [1–8]. Although the experiments with uncouplers on artificial membranes, mitochondria and bacteria have played a crucial role in the validation of Mitchell's chemiosmotic theory, the mechanism of uncoupling remains not fully understood. There is so far no consensus whether the uncoupling action occurs with or without the aid of membrane proteins [9], despite the general confidence in the protonophoric nature of this action.

**Data Availability Statement:** All relevant data are within the manuscript and its Supporting Information files.

**Funding:** This work was financially supported by the Russian Science Foundation, grant number 16-14-10025.

**Competing interests:** The authors have declared that no competing interests exist.

**Abbreviations:** FCCP, carbonyl cyanide 4-(trifluoromethoxy)phenylhydrazone; CCCP, carbonyl cyanide 3-chlorophenylhydrazone; SF6847 tyrphostin A9, 3,5-di-tert-butyl-4-hydroxybenzylidenemalononitrile; DNP, 2,4-dinitrophenol; BAM15, N5,N6-bis(2-fluorophenyl)[1,2,5]oxadiazolo[3,4-b]pyrazine-5,6-diamine; mitoCCCP, 10-([4'-(dicyanomethylene)hydrazinyl-2'-chlorophenyl]oxy)decyl(triphenyl)phosphonium bromide; TPP, triphenylphosphonium; DPhPC, diphytanoylphosphatidylcholine; POPC, 1-palmitoyl-2-oleoyl-sn-glycero-3-phosphocholine; POPG, 1-palmitoyl-2-oleoyl-sn-glycero-3-phosphoglycerol; EggPC, egg yolk phosphatidylcholine; RLM, rat liver mitochondria; BLM, bilayer lipid membrane; ΔΨ, mitochondrial membrane potential; SMP, submitochondrial particles; ACMA, 9-amino-6-chloro-2-methoxyacridine.

In particular, the recoupling effect of 6-ketocholestanol (kCh) found earlier by Starkov and colleagues [10, 11] was tentatively considered as an evidence in favor of involvement of proteins in mitochondrial uncoupling mediated by such agents as carbonyl cyanide-*m*-chlorophenylhydrazone (CCCP), carbonyl cyanide *p*-trifluoromethoxy phenylhydrazone (FCCP) and tyrphostin A9 (SF6847) [9]. This idea was supported by the fact that in planar bilayer lipid membranes, kCh not only failed to reverse the protonophoric action of SF6847, but even enhanced the conductivity increase caused by this uncoupler. Nevertheless, it cannot be excluded that the recoupling action of kCh is somehow associated with its impact on physical properties of membranes, i.e. an increase in the membrane dipole potential [11, 12] and diminution of membrane fluidity [13]. Later, the kCh-induced recoupling was also reported for curcumin derivatives [14], eudesman [15], SR4 [16], a bicyclic hydroquinone [17] and BAM15 [18]. By contrast, kCh was ineffective with 2,4-dinitrophenol (DNP), fatty acids and gramicidin A [11].

To make the impact of uncouplers more selective, i.e. to avoid adverse effects, it is reasonable to design mitochondria-targeted agents. Earlier in the laboratory of Michael Murphy, an attempt was made to obtain a mitochondria-targeted uncoupler based on a conjugate of the popular uncoupler DNP with a triphenylphosphonium (TPP) cation. It was assumed that the addition of a lipophilic cation will provide selective accumulation of the uncoupler in the mitochondria, since energized mitochondria are negatively charged with respect to the cytosol. However, the resulting compound lacked the uncoupling activity [19]. Here, we report synthesis of a mitochondria-targeted derivative of CCCP (a conjugate with TPP coined mitoCCCP), which appeared to be a rather weak uncoupler but exhibited effective inhibition of the CCCP-caused uncoupling in mitochondria.

## 2. Materials and methods

### 2.1. Chemicals

Most chemicals, including CCCP, FCCP, DNP, tyrphostin A9 (3,5-di-tert-butyl-4-hydroxybenzylidenemalononitrile), niclosamide, rotenone, lasalocid A, diphytanoylphosphatidylcholine (DPhPC), and safranine O were from Sigma.

### 2.2. Isolation of rat liver mitochondria and mitoplasts

Mitochondria were isolated from rat liver by using differential centrifugation [20], according to a slightly modified procedure previously described [21]. The animals were handled and experiments were performed in accordance with the international guidelines for animal care and use and the Institutional Ethics Committee of A.N. Belozersky Institute of Physico-Chemical Biology at the Lomonosov Moscow State University approved them (protocol #3 on February 12, 2018). To prepare mitoplasts (mitochondria minus outer membrane), freshly purified mitochondria were resuspended in 10 mM-Tris/HCl buffer, pH 7.5 (swelling medium), at a protein concentration of about 10 mg/ml before incubation on ice for 5 min. The mitoplasts were centrifuged and resuspended in the isolation medium and washed twice in the same buffer.

### 2.3. Preparation of submitochondrial particles from bovine heart

Inside-out coupled SMP of bovine heart mitochondria were prepared and activated as described Bovine heart inside-out submitochondrial particles (SMP) were prepared, activated, and coupled by treatment with oligomycin (0.5 µg/mg of SMP protein), as described in [22].

## 2.4. Preparation of subbacterial inverted membrane particles (SBP) from *Escherichia coli*

Subbacterial inverted membrane particles (SBP) from *Escherichia coli* strain BW25113 were provided by Anna Lapashina (Belozersky Institute, Moscow State University). The particles were prepared as described in [23]. Briefly, after growth cells were washed once with buffer containing 10 mM HEPES-NaOH, pH 7.5, 5 mM MgCl2, and 10% glycerol, resuspended in 25–30 ml of the same buffer, and disrupted by two consecutive passages through a French cell press (SLM Aminco, USA) at 1000 PSI. Unbroken cell debris was collected by 30 min centrifugation at 13,700g and discarded. The supernatant was centrifuged for 1 h at 390,000g, and the SBP pellet was washed with 25 ml of the same buffer and resuspended in the same buffer. The centrifugation was repeated once more, and the SBP pellet was suspended in 1.0–1.5 ml of the same buffer, aliquoted by 50 μl, frozen in liquid nitrogen, and stored at –80˚C.

## 2.5. Mitochondrial respiration

The respiration of isolated rat liver mitochondria was measured at the mitochondrial protein concentration of 0.8 mg/ml by using a Clark-type oxygen electrode (Strathkelvin Instruments, UK), as described previously [21]. The ADP/O ratio was calculated as in [24].

## 2.6. Membrane potential ($\Delta\Psi$) measurement in isolated mitochondria

The mitochondrial membrane potential ($\Delta\Psi$) was evaluated from the difference in the absorbance at 555 and 523 nm ($\Delta A$) of the safranine O dye [25] measured with an Aminco DW-2000 spectrophotometer, as described previously [21]. Mitochondria were incubated in the medium containing 250 mM sucrose, 5 mM MOPS, 0.5 mM $KH_2PO_4$, 1 mM EGTA, 2 μM rotenone, 5 mM succinate (pH 7.4), 1 μg/ml oligomycin, and 15 μM safranine O at the mitochondrial protein content of 0.7–0.9 mg protein/ml.

## 2.7. NADH oxidation by submitochondrial particles

NADH oxidase activity of SMP was measured at 340 nm in the mixture composed of 0.25 M sucrose, 50 mM Tris/HCl (pH 8.0), 0.2 mM EDTA at 30˚C supplemented by 100 μM NADH. The reaction was initiated by the addition of SMP (10 μg of protein per ml).

## 2.8. Generation of pH gradient ($\Delta pH$) in submitochondrial particles

Generation of the transmembrane pH gradient in response to NADH addition was measured by following the changes in fluorescence of 9-amino-6-chloro-2-methoxyacridine (ACMA) with a Spectrofluorometer Fluorat-02-Panorama—Lumex Instruments (410 nm emission, 480 nm excitation), as described in [26]. The buffer solution (pH 7.4) contained 10 mM HEPES, 100 mM KCl, 5 mM $MgCl_2$, 0.5 μg/ml ACMA.

## 2.9. Electrical current through planar bilayers

BLM was formed by the brush technique [27] from 2% diphytanoylphosphatidylcholine (DPhPC) solution in decane on a 0.8 mm aperture in a Teflon septum separating the chamber into two equal compartments (3 ml each). Electrical currents under voltage-clamp conditions were measured with two Ag/AgCl electrodes placed into solutions on each side of BLM via agar bridges (the ground electrode in the *cis* compartment) using a Keithley 428 amplifier (Keithley Instruments, Cleveland, OH). Transmembrane electrical potentials were measured using Keithley 6517A. All experiments were performed at 23-25˚C.

## 2.10. Detection of proton transport in pyranine-loaded liposomes

The lumenal pH of the liposomes was assayed with pyranine [28] by a slightly modified procedure of [29]. To prepare pyranine-loaded liposomes, lipid (2 mg POPC, 1 mg POPG and 1 mg cholesterol) in a chloroform suspension was dried in a round-bottom flask under a stream of nitrogen. The lipid was then resuspended in buffer (100 mM KCl, 20 mM MES, 20 mM MOPS, 20 mM Tricine titrated with KOH to pH 6.0) containing 0.5 mM pyranine. The suspension was vortexed and then freeze-thawed three times. Unilamellar liposomes were prepared by extrusion through 0.1-µm-pore size Nucleopore polycarbonate membranes using an Avanti Mini-Extruder. The unbound pyranine was then removed by passage through a Sephadex G-50 coarse column equilibrated with the same buffer solution. To measure the rate of pH dissipation in liposomes with lumenal pH 6.0, the liposomes were diluted in solution buffered to pH 8 and supplemented with 2 mM p-xylene-bis-pyridinium bromide to suppress the fluorescence of leaked pyranine. The inner liposomal pH was estimated from the pyranine fluorescence intensity measured at 505 nm upon excitation at 455 nm with the Fluorat-02-Panorama spectrofluorometer. At the end of each recording, 1 µM lasalocid A was added to dissipate the remaining pH gradient. To prevent the formation of $H^+$-diffusion potential, the experiments were carried out in the presence of 10 nM valinomycin.

## 3. Results and discussion

The synthesis of mitoCCCP, 10-([4'-(dicyanomethylene)hydrazinyl-2'-chlorophenyl]oxy) decyl(triphenyl)phosphonium bromide (Fig 1), started with the synthesis of p-hydroxyCCCP via the reaction of 4-amino-2-chlorophenol with sodium nitrite in HCl followed by the reaction with malononitrile. In particular, the amino group was converted into a diazonium salt by a diazotization reaction, and then a functional group responsible for the uncoupling activity of CCCP was formed by the nucleophilic addition of malononitrile. Finally, the 10-bromodecyl (triphenyl)phosphonium cation obtained by quaternization was attached to the hydroxy group by an O-alkylation reaction (Fig 1). ESI-MS data for p-hydroxyCCCP (S1 Fig in S1 File) and mitoCCCP (S2 Fig in S1 File) can be found in ESI.

Fig 2A illustrates an effect of mitoCCCP (added at t = 60 s) on the membrane potential of isolated rat liver mitochondria (RLM), as measured by changes in absorbance of the potential-

**Fig 1. A scheme of the synthesis of mitoCCCP, a derivative of the protonophore CCCP (carbonyl cyanide *m*-chlorophenylhydrazone) targeted to mitochondria by a decyl(triphenyl)phosphonium cation.**

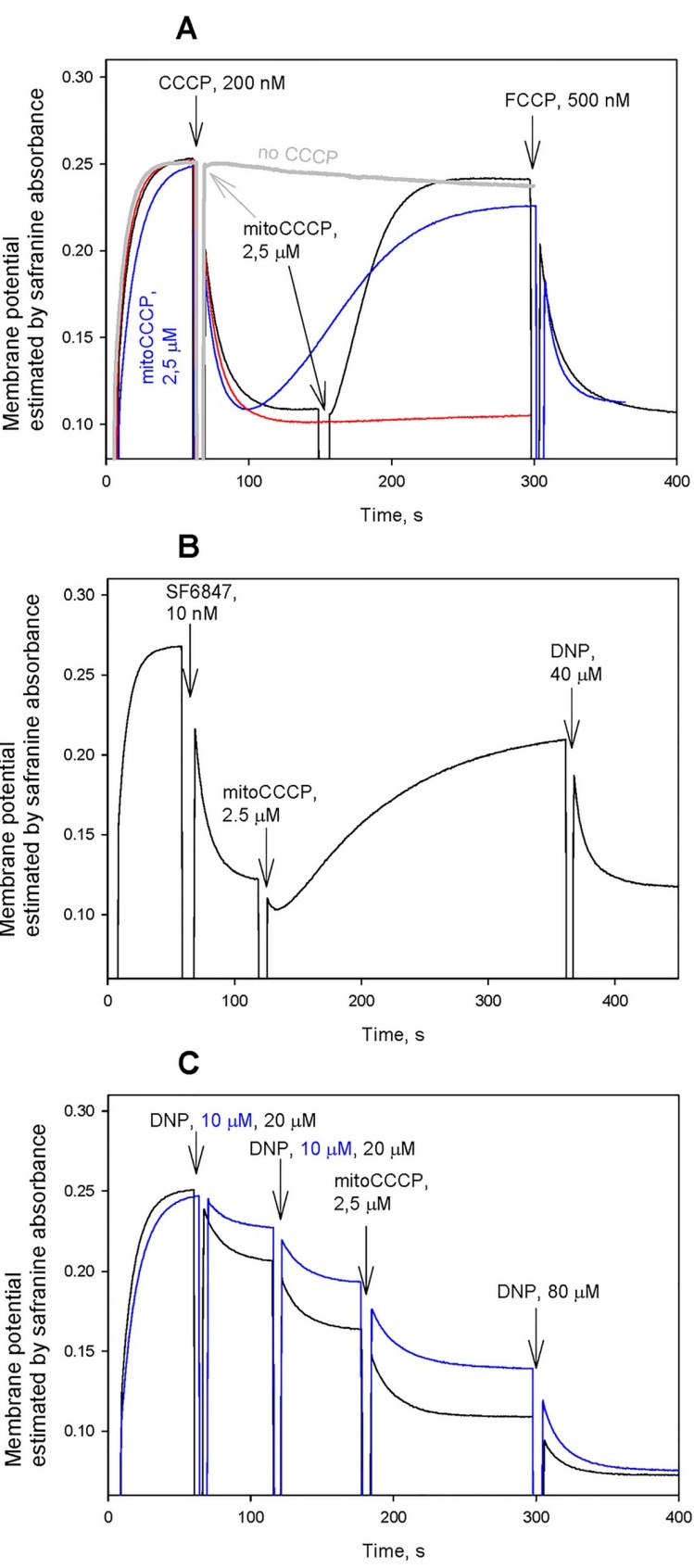

**Fig 2.** Effect of mitoCCCP (2.5 μM) on the uncoupling activity of CCCP (A), SF6847 (B) or DNP (C) in rat liver mitochondria. Substrate: succinate. The membrane potential of mitochondria was estimated from changes in the absorbance of the potential sensitive dye safranine O (15 μM) at 555 nm and 523 nm. For other conditions, see Materials and methods.

sensitive dye safranine O [25]. The addition of 2.5 μM mitoCCCP to RLM at t = 60 s did not lead to a decrease in the membrane potential (gray curve in Fig 2A), in contrast to CCCP (0.2 μM, black curve in Fig 2A). However, the addition of mitoCCCP at t = 150 s after CCCP resulted in restoration of the initial level of the membrane potential observed in the absence of CCCP. The subsequent addition of FCCP at a high concentration (0.5 μM) again brought about a drop in the membrane potential. If mitoCCCP was added prior to CCCP, the depolarizing effect of CCCP became transient and disappeared in 2 minutes (blue curve). The recoupling effect of mitoCCCP qualitatively resembled that of kCh [10, 11], albeit mitoCCCP effectively recoupled RLM at much lower concentrations than kCh. The dose dependence of the recoupling action of mitoCCCP is shown in S3 Fig in S1 File, while S4 Fig in S1 File shows that the recoupling action of 2.5 μM mitoCCCP remains nearly complete in a wide range of CCCP concentrations except for concentrations exceeding 300 nM. Interestingly the recoupling action of mitoCCCP exhibited transient character in case of mitoplasts, i.e. mitochondria lacking the outer membrane (S5 Fig in S1 File). The initial rise of the membrane potential after the addition of mitoCCCP was followed by a slow decrease lasting several minutes. The recoupling by mitoCCCP was also observed if SF6847 (Fig 2B) or niclosamide (data not shown) were used as mitochondrial uncouplers, although the recoupling was lower than in the case of CCCP. However, the effect of mitoCCCP was absolutely different if DNP was used instead of CCCP: the addition of mitoCCCP did not cause restoration of the membrane potential dropped by DNP. On the contrary, the depolarizing action of DNP was enhanced by mitoCCCP (Fig 2C), which was consistent with our previous data on the interaction of anionic protonophores with lipophilic cations such as decyl(triphenyl)phosphonium (C10TPP) and its derivatives [30].

To confirm the inhibitory effect of mitoCCCP on the CCCP-mediated uncoupling of RLM, we performed measurements of mitochondrial respiration, because the uncoupler-induced decrease of mitochondrial membrane potential is known to be accompanied by acceleration of mitochondrial respiration due to the dependence of the turnover rate of respiratory proton pumps on the membrane potential. As seen from Fig 3, blue curve, addition of 100 nM CCCP increased the respiration rate from 12 to 42. Remarkably, the subsequent addition of 2.5 μM mitoCCCP suppressed the respiration rate to 22, which could be again augmented by the addition of ADP or DNP at a high concentration (80 μM), thereby showing that the mitochondrial recoupling by mitoCCCP was not associated with inhibition of the respiratory chain. Similar to the observations on membrane potential, the DNP-induced stimulation of mitochondrial respiration, in contrast to the CCCP-induced uncoupling, was not removed by mitoCCCP (green curve, Fig 2). Insert to Fig 3 displays the effect of CCCP on the ADP/O ratio showing the stoichiometry of ATP synthetized per oxygen consumed. The ADP/O value decreased by CCCP was restored by mitoCCCP, thereby confirming its recoupling action.

Similar to speculation on the mechanism of the recoupling action of 6-ketocholestanol [10, 11], it could be suggested that the recoupling activity of mitoCCCP is mediated by mitochondrial proteins. Alternatively, mitoCCCP, as a lipophilic cation derivative, could form inactive complexes with CCCP without participation of proteins, thereby resulting in a decrease in the CCCP concentration in solution, which could manifest itself as inhibition of the CCCP-mediated uncoupling. To examine this possibility, we studied impact of mitoCCCP on the CCCP-mediated proton current across planar bilayer lipid membrane (BLM) formed by the brush technique [27] (Fig 4A). The addition of mitoCCCP resulted in significant stimulation of the

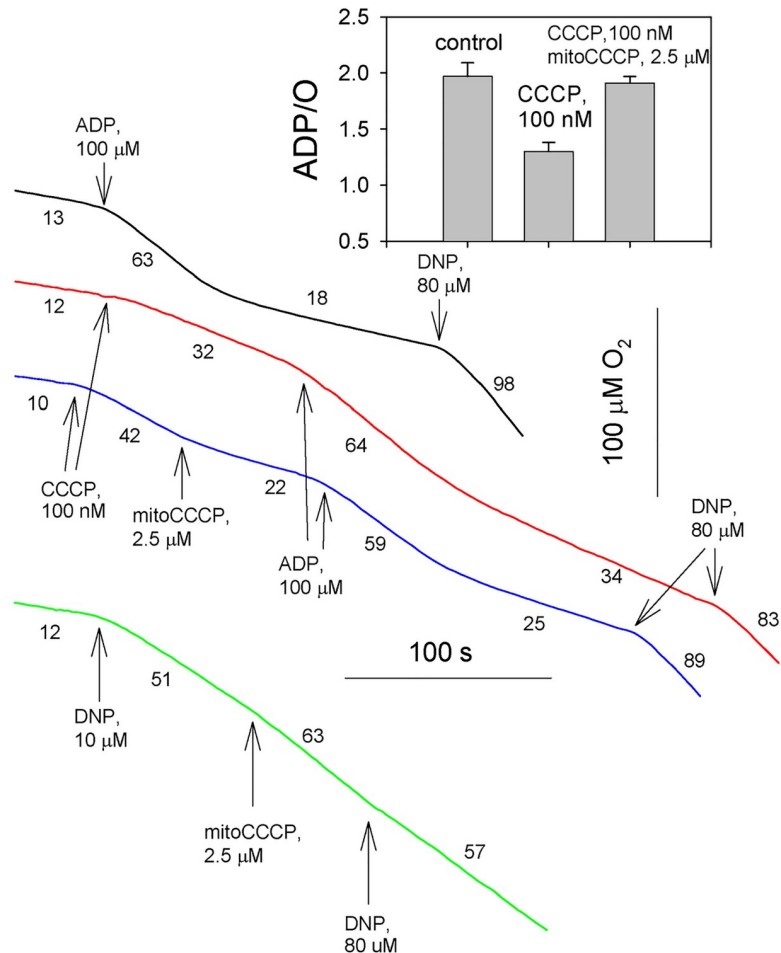

**Fig 3. Effect of mitoCCCP (2.5 μM) on the stimulation of succinate-driven respiration of rat liver mitochondria by CCCP (100 nM, blue curve) or by DNP (10 μM, green curve).** The three top traces show the stimulation of respiration by 100 μM ADP due to ATP synthesis. Insert: Recoupling effect of mitoCCCP on the ADP/O ratio (amount of ADP phosphorylated/O atom consumed). Shown are Mean±S.D. (n = 4). For experimental conditions, see Materials and Methods.

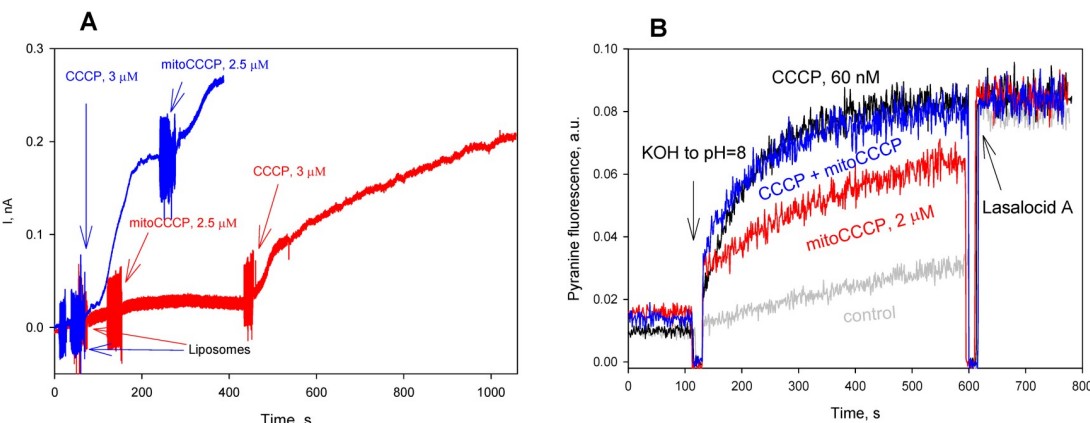

**Fig 4. A**. Effect of mitoCCCP (2.5 μM) on the CCCP (3 μM)—mediated electrical current through planar bilayer lipid membrane (BLM) made from DPhPC. The solution was 50 mM Tris, 50 mM MES, 10 mM KCl, 10 μg/ml EggPC liposomes, pH 7.4. The voltage applied to BLM was 50 mV. **B**. Effect of mitoCCCP (2 μM) on the CCCP (60 nM)–mediated proton fluxes through liposomes loaded with the pH-probe pyranine. Inner liposomal pH was estimated from the pyranine fluorescence intensities measured at 505 nm upon excitation at 455 nm. 1 μM lasalocid A was added at 600 s to equilibrate the pH. Other conditions: see Materials and Methods. Lipid concentration was 20 μg/ml.

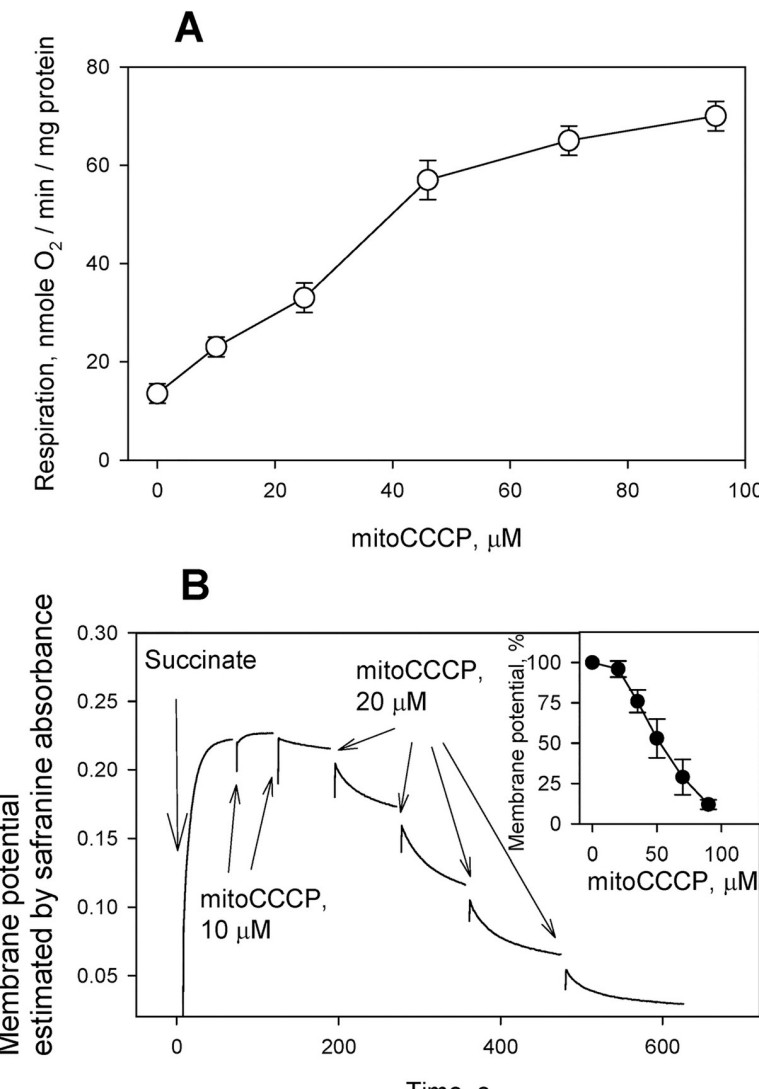

**Fig 5. A**. Dose dependence of respiratory stimulation by mitoCCCP with succinate as a substrate of mitochondria. Shown are Mean±S.D. (n = 4). For experimental conditions, see Materials and methods. **B**. Effect of mitoCCCP on the mitochondrial membrane potential in rat liver mitochondria (RLM) estimated by measuring absorbance changes of the potential-sensitive dye safranine O (15 μM). For experimental conditions, see Materials and methods. Insert: dose dependence of the effect of mitoCCCP on the membrane potential of RLM (Mean±S.D., n = 3).

CCCP-mediated BLM current (blue curve in Fig 4), whereas mitoCCCP by itself did not elicit a noticeable BLM current (red curve in Fig 4), in agreement with the poor uncoupling effect of mitoCCCP on mitochondria. The mitoCCCP-caused stimulation of the CCCP-mediated BLM current was compatible with our previous data on the effect of lipophilic cations on the proton current induced by anionic uncouplers [30]. In addition, we performed experiments on liposomes loaded with the pH-sensitive fluorescent dye pyranine [29]. Fig 4B shows the kinetics of dissipation of pH gradient on the liposomal membranes after the addition of CCCP (60 nM, black curves), mitoCCCP (2 μM, red curve), and their combination (green curve). A slight acceleration of the CCCP-mediated proton flux was seen in the presence of mitoCCCP. Hence, in artificial lipid membrane systems (planar bilayers and liposomes), interaction between CCCP and mitoCCCP could lead only to stimulation of the proton flow across lipid

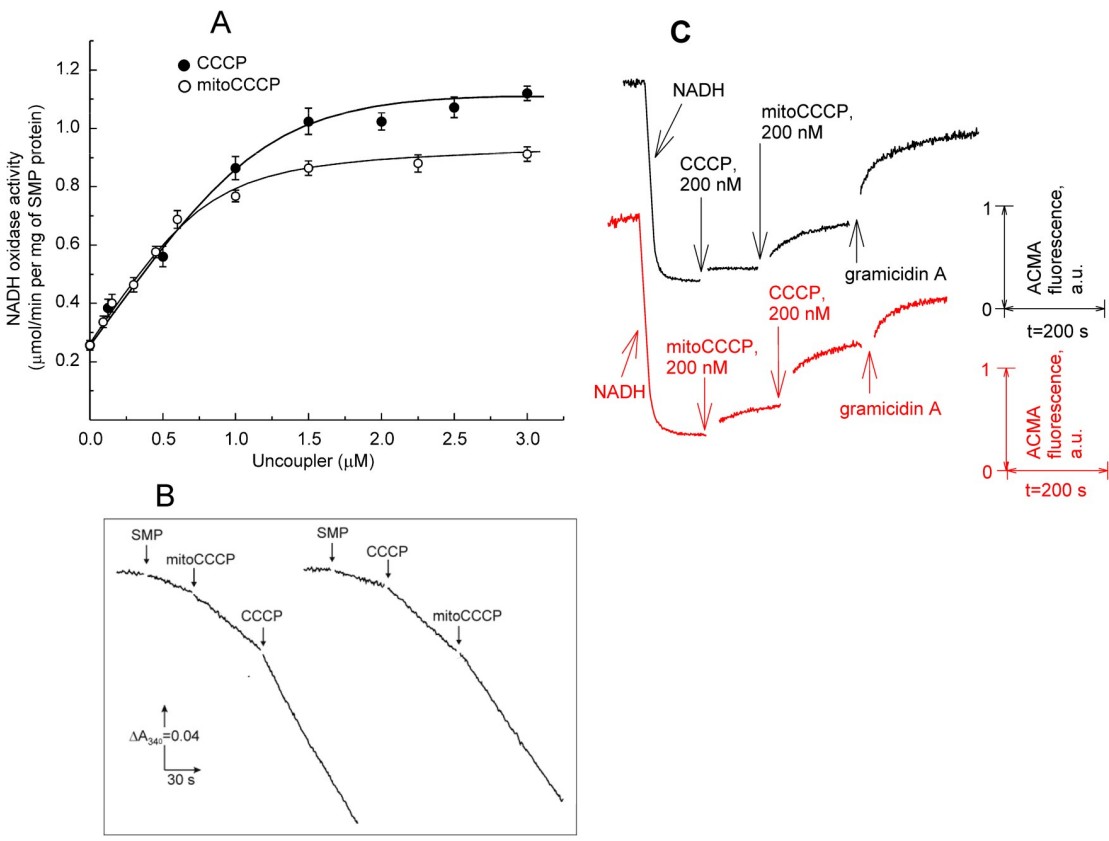

**Fig 6. A**. Concentration dependence of the effect of mitoCCCP (open circles) and CCCP (closed circles) on the NADH oxidation rate in submitochondrial particles (SMP). All the data are presented as mean ± SEM of at least 3 independent experiments. For experimental conditions, see Materials and methods. **B**. Traces of NADH oxidation by SMP in the presence of mitoCCCP and CCCP. Concentrations, left: mitoCCCP 0.45 μM, CCCP, 2.5 μM; right: CCCP, 1 μM, mitoCCCP, 0.45 μM. SMP concentration, 10 μg/ml. **C**. Effect of mitoCCCP and CCCP on the pH gradient across the membranes of submitochondrial particles (SMP), as measured by ACMA (0.5 μg/ml) fluorescence. Shown are traces of fluorescence at 480 nm (excited at 410 nm) in the medium containing 100 mM KCl, 10 mM HEPES, 5 mM $MgCl_2$, 0.5 mM EGTA, pH 7.4. In the traces, 200 μM NADH was supplemented at t = 150 s. 1 μg/ml of gramicidin A was added at the end of each trace. Protein concentration was 25 μg/ml.

membranes, but not to suppression of the protonophoric activity of CCCP, as it was observed with mitochondria (Fig 2A).

Of note, when added at high concentrations (tens of micromoles), mitoCCCP caused mitochondrial uncoupling (decreased membrane potential and stimulated respiration, Fig 5), thereby manifesting itself as an uncoupler of a zwitterionic type, similar to protonophores described in [31–33]. The zwitterionic protonophores are supposed to cross the membrane as a zwitterion in the deprotonated form and as a cation in the protonated form [33]. The weakness of the uncoupling activity of mitoCCCP in mitochondria might be associated with a much lower value of its pKa, 3.7 (S6 Fig in S1 File) compared to that of CCCP (5.95) [34, 35]. Surprisingly, with submitochondrial particles (SMP) isolated from bovine heart, mitoCCCP exhibited the pronounced uncoupling activity, as measured by both the NADH oxidation (Fig 6A and 6B) and the fluorescence response of the pH gradient-sensitive dye ACMA (Fig 6C). Fig 6A shows dependences of the rate of NADH oxidation by the respiratory electron transfer chain on the concentrations of mitoCCCP and CCCP. The concentration corresponding to half-maximum stimulation of respiration was about 400 nM for both compounds. The addition of mitoCCCP not only did not decelerate, but noticeably accelerate the NADH oxidation

stimulated by CCCP (Fig 6B, right trace). Moreover, the pH gradient generated on SMP after the addition of NADH, manifesting itself in the quenching of ACMA fluorescence [26], was substantially decreased by 200 nM mitoCCCP (Fig 6C, red curve). The addition of 200 nM mitoCCCP after 200 nM CCCP led to an additional decrease in the pH gradient (black curve). Therefore, no recoupling was observed with mitoCCCP in SMP, in contrast to the case of mitochondria. In addition, we performed experiments with subbacterial particles from *E. coli*. The results were similar to those obtained with SMP, i.e. submicromolar concentrations of mitoCCCP induced depolarization of these vesicles and the addition of mitoCCCP after CCCP led to further reduction of the pH gradient (S7 Fig in S1 File). The effective concentrations of mitoCCCP in subbacterial particles were similar to those in SMP, i.e. hundreds of nanomolars.

Thus, by conjugating the classical uncoupler CCCP with the lipophilic cation TPP, we obtained the compound that can block the CCCP-mediated uncoupling in mitochondria but not in SMP, and does not reduce the protonophoric activity of CCCP in protein-free lipid membranes, which may indicate participation of certain proteins in the mitochondrial uncoupling, as previously suggested [11]. This suggestion is compatible with the data on CCCP binding to mitochondrial proteins [36]. In particular, CCCP was shown to have high-affinity binding sites on cytochrome $c$ oxidase, $bc_1$ complex and $H^+$-ATPase [37, 38], but whether these interactions contribute to the uncoupling was debated [9]. It can be speculated that mitoCCCP can bind to the same binding site/sites as CCCP, thereby preventing the protein-assisted protonophoric uncoupling mediated by CCCP, while mitoCCCP is a much weaker protonophore than CCCP according to our data.

## Supporting information

**S1 File.**
(DOC)

## Acknowledgments

We are thankful to Elena Y. Pyrkova for technical assistance, and Anna S. Lapashina for providing subbacterial particles.

## Author Contributions

**Conceptualization:** Yuri N. Antonenko.

**Data curation:** Yuri N. Antonenko.

**Formal analysis:** Yuri N. Antonenko.

**Funding acquisition:** Yuri N. Antonenko.

**Investigation:** Iliuza R. Iaubasarova, Ljudmila S. Khailova, Alexander M. Firsov, Vera G. Grivennikova, Roman S. Kirsanov, Yuri N. Antonenko.

**Methodology:** Galina A. Korshunova, Yuri N. Antonenko.

**Project administration:** Yuri N. Antonenko.

**Resources:** Yuri N. Antonenko.

**Supervision:** Yuri N. Antonenko.

**Validation:** Elena A. Kotova, Yuri N. Antonenko.

**Writing – original draft:** Elena A. Kotova, Yuri N. Antonenko.

**Writing – review & editing:** Yuri N. Antonenko.

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
