## [Decision Letter · Decision Letter 0]

17 Jul 2020

PONE-D-20-17539

The mitochondria-targeted derivative of the classical uncoupler of oxidative phosphorylation carbonyl cyanide m-chlorophenylhydrazone is an effective mitochondrial recoupler

PLOS ONE

Dear Dr. Antonenko,

Thank you for submitting your manuscript to PLOS ONE.  Both reviewers and the editor concur that the findings reported in your paper merit publication, but several important points need to be addressed before the paper can be accepted. Therefore, we invite you to submit a revised version of the manuscript including some additional data and changes in response to the comments of the reviewers.

As noted by Reviewer #1, it is essential to examine the effect of mito-CCCP on the uncoupling activity of CCCP in mitoplasts,  and mitochondrial inner membrane  vesicles. Dependence of the recoupling effect of mito-CCCP on CCCP concentration, in mitochondria, and in these other membrane preparations, if recoupling is observed in the latter, should also be included in the revised manuscript. Please also address in the discussion  comments of Reviewer #2 regarding comparison between the effects of 6-ketocholestanol and mito-CCCP. Please respond to all other comments of the reviewers appended below.

We look forward to receiving your revised manuscript.

Kind regards,

Oleg Y. Dmitriev, Ph.D.

Academic Editor

PLOS ONE

Journal Requirements:

2. In line with our guidelines on animal reporting please include the following:

•        Please specify the source from which animals were obtained

•        Please specify method of anaesthesia used.

•        Please specify the method of euthanasia.

•        Please complete and submit a copy of the ARRIVE Guidelines checklist, a document that aims to improve experimental reporting and reproducibility of animal studies for purposes of post-publication data analysis and reproducibility: https://www.nc3rs.org.uk/arrive-guidelines.

Please include your completed checklist as a Supporting Information file. Note that if your paper is accepted for publication, this checklist will be published as part of your article."

'This work was financially supported by the Russian Science Foundation, grant number 16-14-10025...'

'No'

5. Thank you for stating the following in your Competing Interests section: 'No'

a. Please complete your Competing Interests statement to state any Competing Interests.

If you have no competing interests, please state "The authors have declared that no competing interests exist.", as detailed online in our guide for authors at http://journals.plos.org/plosone/s/submit-now

7. PLOS requires an ORCID iD for the corresponding author in Editorial Manager on papers submitted after December 6th, 2016. Please ensure that you have an ORCID iD and that it is validated in Editorial Manager. To do this, go to ‘Update my Information’ (in the upper left-hand corner of the main menu), and click on the Fetch/Validate link next to the ORCID field. This will take you to the ORCID site and allow you to create a new iD or authenticate a pre-existing iD in Editorial Manager. Please see the following video for instructions on linking an ORCID iD to your Editorial Manager account: https://www.youtube.com/watch?v=_xcclfuvtxQ

8. Please include captions for your Supporting Information files at the end of your manuscript, and update any in-text citations to match accordingly. Please see our Supporting Information guidelines for more information: http://journals.plos.org/plosone/s/supporting-information

Reviewers' comments:

Reviewer's Responses to Questions

**Comments to the Author**

1. Is the manuscript technically sound, and do the data support the conclusions?

Reviewer #1: Yes

Reviewer #2: Partly

2. Has the statistical analysis been performed appropriately and rigorously? 

Reviewer #1: N/A

Reviewer #2: I Don't Know

3. Have the authors made all data underlying the findings in their manuscript fully available?

Reviewer #1: Yes

Reviewer #2: Yes

4. Is the manuscript presented in an intelligible fashion and written in standard English?

Reviewer #1: Yes

Reviewer #2: Yes

5. Review Comments to the Author

Reviewer #1: This is a provocative paper that reports new data that suggests that the classical uncoupler CCCP operates in rat liver mitochondria by a mechanism that is distinct from its protonophore function. A derivative of CCCP, mito-CCCP has been synthsized and shown to effectively "recouple" mitochondria that have been decoupled by CCCP. The conclusion is that a protein-mediated mechanism of CCCP uncoupling should be considered. Protonophore activity of CCCP in both planar membanes and liposomes was not inhibited by mito-CCCP, nor was the uncoupling of mitochondria by DNP. The authors consider a direct complex formation between CCCP and mito-CCCP and conclude that this cannot be the explanation of the observed inhibition because the mito-CCCP accelerates the protonophore activity in model membrane systems. However, these data suggest there may be a direct interaction between the two compounds although a dramatic difference between model membranes and mitochondria must still be concluded. As far as it goes, this is a solid study and should be published. I suggest that the paper would be more complete if the authors also examined mitoplasts (inverted vesicles) and inverted E. coli vesicles to explore whether their observations apply to other biological membranes or inverted membranes. Since CCCP is clearly an potent protonophore the question is why this itself is not sufficient to result in uncoupling with mitochondria even if another protein-based mechanism is also operative. Concentration-dependence of CCCP on the ability to recouple by mito-CCCP might be instructive.

Reviewer #2: The authors have developed an interesting chemical tool; however, the mechanism remains incompletely understood and the paper would benefit from benchmarking to another recoupler.

1. Does mito-CCCP affect the lipid membrane dipole moment?

2. The authors should directly compare mito-CCCP to 6-ketocholestanol to better understand the difference between these recouplers. i.e. repeat figure 2A-C but with 6-ketocholestanol instead of mitoCCCP.

3. 6-KCh alters membrane dipole moment and this can block anion permeation alter hydration and phase behaviour of the membrane. Whether or not 6Kch blocks an uncoupler does not definitively mean that proteins are involved and the authors should thoroughly explain all possibilities. The authors only have 1 sentence on 6-Kch in the intro, which should be expanded into at least one paragraph that highlights what is known about this recoupler.

4. Does 6-KCh alter the activity of mito-CCCP over a dose-response of mitoCCCP?

5. Figures 2-3 use isolated mitochondria that can regenerate a proton gradient, while Fig 4B inovlves liposomes that cannot so there are multiple possibilities other than involvement of protein that may explain differences in compound activity in mitos vs liposomes. The authors should repeat figure 2A in the presence of non-respiring mitochondria treated with antimycin A and rotenone.

6. PLOS authors have the option to publish the peer review history of their article (what does this mean?). If published, this will include your full peer review and any attached files.

Reviewer #1: No

Reviewer #2: No

---

## [Author Response · Author response to Decision Letter 0]

14 Oct 2020

A.N.Belozersky Institute of Physico-Chemical Biology

 Moscow State University

 tel. +74-95-939-5149

 fax +74-95-939-3181

Editorial Office

PLOS ONE

 Moscow, 09.10.2020

Dear Professor Dmitriev, 

thank you for reviewing our manuscript. According to your suggestions, we ran additional experiments on mitoplasts and sub-mitochondrial particles (SMP) as well as on sub-bacterial particles (SBP). While we observed similar recoupling effect of mitoCCCP in mitoplasts as in mitochondria, we did not observe the recoupling effect in case of membrane particles. Moreover, in contrast to mitochondria, mitoCCCP turns out to uncouple the particles at sub-micromolar concentrations (Fig.A below). We examined the dependence of the mitoCCCP recoupling on the concentration of CCCP and included it in the revised version. We feel that the effect of mitoCCCP on SMP/SBP is beyond the scope of our manuscript and we would like to publish these data in a separate paper along with the effect of other cationic conjugates of TPP with fluorescein and dinitrophenol. 

We are grateful for critical evaluation of our manuscript by the reviewers. Please, find below our responses. 

Reviewer #1 

Comment.

This is a provocative paper that reports new data that suggests that the classical uncoupler CCCP operates in rat liver mitochondria by a mechanism that is distinct from its protonophore function. A derivative of CCCP, mito-CCCP has been synthsized and shown to effectively "recouple" mitochondria that have been decoupled by CCCP. The conclusion is that a protein-mediated mechanism of CCCP uncoupling should be considered. Protonophore activity of CCCP in both planar membanes and liposomes was not inhibited by mito-CCCP, nor was the uncoupling of mitochondria by DNP. The authors consider a direct complex formation between CCCP and mito-CCCP and conclude that this cannot be the explanation of the observed inhibition because the mito-CCCP accelerates the protonophore activity in model membrane systems. However, these data suggest there may be a direct interaction between the two compounds although a dramatic difference between model membranes and mitochondria must still be concluded. As far as it goes, this is a solid study and should be published.

Answer

Not required

Comment.

I suggest that the paper would be more complete if the authors also examined mitoplasts (inverted vesicles) and inverted E. coli vesicles to explore whether their observations apply to other biological membranes or inverted membranes. 

Answer

We prepared mitoplasts and examined the effect of mitoCCCP on the CCCP-mediated decrease in the membrane potential. The effect was similar to that in mitochondria except the apparent transient character of the recoupling (Fig.S5 of the revised version). We speculate that the slow disappearance of the effect of mitoCCCP in case of mitoplasts could be accounted for by the limited tightness of the inner mitochondrial membrane induced by a stage of high-amplitude swelling of mitochondria during the procedure of the preparation of mitoplasts. In order to test the effect on the inverted vesicles of mitochondria, we ran experiments on sub-mitochondrial particles (SMP) from bovine heart provided by Dr.Vera Grivennikova (a Lab of Professor Vinogradov, Department of Biology, Moscow State University). Figure A below shows that the pH gradient generated on SMP after the addition of NADH can be substantially decreased by 60 nM mitoCCCP (red curve), the effect was similar to that of 200 nM CCCP (black curve). The addition of 60 nM mitoCCCP after 200 nM CCCP led to an additional decrease in the pH gradient (black curve). The effect of mitoCCCP cannot be attributed to inhibition of the respiratory chain of the SMP because the measurements of the rate of NADH consumption by absorbance at 340 nm shows a stimulation of the electron flow by this concentration of mitoCCCP (data not shown). As written above, we think that description of the strong uncoupling effect of mitoCCCP on SMP/SBP is beyond the scope of our manuscript and we would like to publish these data in a separate paper along with the effect of other cationic conjugates of TPP with fluorescein and dinitrophenol.

Fig. A. Effect of mitoCCCP and CCCP on the pH gradient on the membranes of sub-mitochondrial particles (SMP) measured by ACMA (0.5 µg/ml) fluorescence. Shown are traces of fluorescence in the medium containing 100 mM KCl, 10 mM HEPES, 5 mM MgCl2, 0.5 mM EGTA, pH 7.4. In the traces, 100 µM NADH was supplemented at t=180 s. Protein concentration was 25 µg/ml.

. 

In addition to that we performed experiments with sub-bacterial particles (from E.coli) provided by Anna Lapashina (a Lab of Dr. Boris Feniouk, Department of Bioengeering and Bioinformatics, Moscow State University). The results were similar to those obtained with SMP, i.e. submicromolar concentrations of mitoCCCP induced depolarization of these vesicles and the addition of mitoCCCP after CCCP led to further reduction of the pH gradient. 

Comment.

Since CCCP is clearly an potent protonophore the question is why this itself is not sufficient to result in uncoupling with mitochondria even if another protein-based mechanism is also operative. 

Answer

This is a good question. The answer may be that the protonophoriс properties of CCCP on the lipid membrane are actually not so strong. It has been shown recently in our laboratory that triclosan exhibits much stronger protonophoric action on lipid membranes than CCCP while the uncoupling action of triclosan on mitochondria is substantially weaker than CCCP (Popova et al., BBA, 2018;1860(5):1000-1007. PMID: 29317196). This may indicate that the protonophoric action of CCCP by itself (I mean the action through lipid part of the inner mitochondrial membrane) is actually rather weak and it conducts protons predominantly via some mitochondrial membrane protein. In line with this we have found that the action of CCCP on mitochondria is strongly inhibited by bicarbonate anions (Khailova et al., BBRC 2020;530(1):29-34. PMID: 32828301). 

Comment.

Concentration-dependence of CCCP on the ability to recouple by mito-CCCP might be instructive.

Answer

We measured the dependence of the effect of mitoCCCP on the concentration of CCCP and added these data in the revised version of our manuscript (Figure S4). 

Reviewer #2

1. Comment.

Does mito-CCCP affect the lipid membrane dipole moment? 

Answer

The answer to this question would require substantial piece of work using monolayer technique as well as planar bilayers. However as we wrote in the answer to comment #3 of the reviewer #2, this work is hardly relevant to our study. 

2. Comment.

The authors should directly compare mito-CCCP to 6-ketocholestanol to better understand the difference between these recouplers. i.e. repeat figure 2A-C but with 6-ketocholestanol instead of mitoCCCP. 

Answer

We confirmed the recoupling effect of 6-KCh in case of CCCP (as well as SF6847) in our experiments on mitochondria. Besides we can confirm that 6-KCh does not recouple in case of dinitrophenol. An example of the recoupling action of 6-KCh in case of FCCP has been published by us in (Antonenko et al., BBA 2016; 1860:2463-2473). Because the experiments proposed by the reviewer #2 would not show any new findings we prefer not to include them in our manuscript. 

3. Comment.

6-KCh alters membrane dipole moment and this can block anion permeation alter hydration and phase behaviour of the membrane. Whether or not 6Kch blocks an uncoupler does not definitively mean that proteins are involved and the authors should thoroughly explain all possibilities. The authors only have 1 sentence on 6-Kch in the intro, which should be expanded into at least one paragraph that highlights what is known about this recoupler. 

Answer

We added more information about 6-KCh in the introduction as suggested. In this regard, we would like to emphasize that in our opinion, the effect of 6-KCh on mitochondria is rather mysterious than understandable. The reviewer wrote that the major action of 6-KCh is the alteration of the dipole potential of the membrane which “can block anion permeation”. This is true with respect to dipole potential but definitely not true with respect to anion permeation. 6-Kch actually increases the dipole potential (Franklin and Cafiso, 1993, Biophys J., 65(1):289-299, PMID: 8396456) which increases the anion permeation in planar lipid bilayers. For example, it has been shown by Starkov et al. that 6-Kch increased the SF6847-mediated electrical current in BLM (Table 3 in Biochim Biophys Acta, 1997, 1318(1-2):159-172, PMID: 9030261). Therefore 6-KCh exerts the recoupling action (of say CCCP) on mitochondria in spite of facilitating anion permeation! 

4. Comment.

Does 6-KCh alter the activity of mito-CCCP over a dose-response of mitoCCCP? 

Answer

We tried to test the interference of 6-KCh and mitoCCCP if added one after another in experiments with membrane potential of RLM. Unfortunately we were not able to get a clear answer about the action of 6-KCh on the recoupling effect of mitoCCCP or vise versa. This question requires substantial efforts and we would like to leave it to future work. 

5. Comment.

Figures 2-3 use isolated mitochondria that can regenerate a proton gradient, while Fig 4B inovlves liposomes that cannot so there are multiple possibilities other than involvement of protein that may explain differences in compound activity in mitos vs liposomes. The authors should repeat figure 2A in the presence of non-respiring mitochondria treated with antimycin A and rotenone. 

Answer

Figure B below shows the data of an experiment as suggested by the reviewer #2. In this experiment a small amount of antimycin A was added after reaching the steady-state value of membrane potential at t=1 min. At t=3 min and t=4 min two portions of untreated RLM were added leading to an apparent rise in the membrane potential. Next we added 200 nM CCCP and 3 μM mitoCCCP. The recoupling effect of mitoCCCP is clearly seen in Figure B under these conditions. We think however that this result is hard to interpret and it does not clarify the mechanism of the recoupling action of mitoCCCP. 

Fig. B. Effect of mitoCCCP (3 µM) on the uncoupling activity of CCCP in the presence of rat liver mitochondria treated with antimycin A. Substrate: succinate. At t=190 s and 260 s two additions of RLM (0.8 mg/ml) were supplemented. The membrane potential of mitochondria was estimated from changes in the absorbance of the potential sensitive dye safranine O (15 µM) at 555 nm and 523 nm. For other conditions, see Materials and methods.

 Sincerely yours,

 Yuri Antonenko 

---

## [Decision Letter · Decision Letter 1]

2 Nov 2020

PONE-D-20-17539R1

The mitochondria-targeted derivative of the classical uncoupler of oxidative phosphorylation carbonyl cyanide m-chlorophenylhydrazone is an effective mitochondrial recoupler

PLOS ONE

Dear Dr. Antonenko,

Thank you for submitting your revised manuscript to PLOS ONE. While the authors have performed the additional experiments requested by the reviewers, they chose not to include the results in the revised manuscript. In my opinion, these results are essential for interpreting the "recoupling" effect of mito-CCCP and cannot be reasonably omitted from the paper. I would like to offer you another opportunity to revise your manuscript and include the results of investigating mito-CCCP activity in the membrane vesicles from mitochondria and bacteria. Without this data, I will have no choice but to reject the manuscript.  Considering the significance of the new data, the revised manuscript may have to undergo a full review. 

We look forward to receiving your revised manuscript.

Kind regards,

Oleg Y. Dmitriev, Ph.D.

Academic Editor

PLOS ONE

Reviewers' comments:

Reviewer's Responses to Questions

**Comments to the Author**

1. If the authors have adequately addressed your comments raised in a previous round of review and you feel that this manuscript is now acceptable for publication, you may indicate that here to bypass the “Comments to the Author” section, enter your conflict of interest statement in the “Confidential to Editor” section, and submit your "Accept" recommendation.

Reviewer #2: (No Response)

2. Is the manuscript technically sound, and do the data support the conclusions?

Reviewer #2: Partly

3. Has the statistical analysis been performed appropriately and rigorously? 

Reviewer #2: I Don't Know

4. Have the authors made all data underlying the findings in their manuscript fully available?

Reviewer #2: No

5. Is the manuscript presented in an intelligible fashion and written in standard English?

Reviewer #2: Yes

6. Review Comments to the Author

Reviewer #2: The authors describe a new compound that has recoupling properties. There are few other recouplers demonstrated in the literature so this paper is of interest. However, the authors incompletely characterize their molecule in terms of experiments and dose responses and so this paper is incomplete. The authors state they have some data that they want to save for another paper or don't want to show. It will be hard for other groups to use this molecule because this story is not complete and the authors have refused additional experiments necessary to understand it.

7. PLOS authors have the option to publish the peer review history of their article (what does this mean?). If published, this will include your full peer review and any attached files.

Reviewer #2: No

---

## [Author Response · Author response to Decision Letter 1]

23 Nov 2020

A.N.Belozersky Institute of Physico-Chemical Biology

 Moscow State University

 tel. +74-95-939-5149

 fax +74-95-939-3181

Editorial Office

PLOS ONE

 Moscow, 19.11.2020

Dear Professor Dmitriev, 

According to your suggestion, we ran additional experiments on submitochondrial particles (SMP) and subbacterial particles (SBP) and included the results in the revised manuscript (new Fig.6 and Fig.S7 of ESI). We included Dr. Vera Grivennikova who ran most of the SMP experiments in the list of authors. We hope that the revised manuscript is now acceptable for publication in PLOS ONE.

Reviewer #2 

Comment.

The authors describe a new compound that has recoupling properties. There are few other recouplers demonstrated in the literature so this paper is of interest. However, the authors incompletely characterize their molecule in terms of experiments and dose responses and so this paper is incomplete. The authors state they have some data that they want to save for another paper or don't want to show. It will be hard for other groups to use this molecule because this story is not complete and the authors have refused additional experiments necessary to understand it.

Answer

According to the suggestion, we ran additional experiments on submitochondrial particles (SMP) and subbacterial particles (SBP) and included the results in the revised manuscript (new Fig.6 and Fig.S7 of ESI).

 Sincerely yours,

 Yuri Antonenko 

---

## [Editor Report · Decision Letter 2]

11 Dec 2020

The mitochondria-targeted derivative of the classical uncoupler of oxidative phosphorylation carbonyl cyanide m-chlorophenylhydrazone is an effective mitochondrial recoupler

PONE-D-20-17539R2

Dear Dr. Antonenko,

We’re pleased to inform you that your manuscript has been judged scientifically suitable for publication and will be formally accepted for publication once it meets all outstanding technical requirements. 

IMPORTANT: Please check all figures and make sure that dot and not comma is used as decimal separator.

Kind regards,

Oleg Y. Dmitriev, Ph.D.

Academic Editor

PLOS ONE
---

## [Editor Report · Acceptance letter]

18 Dec 2020

PONE-D-20-17539R2 

The mitochondria-targeted derivative of the classical uncoupler of oxidative phosphorylation carbonyl cyanide *m*-chlorophenylhydrazone is an effective mitochondrial recoupler 

Dear Dr. Antonenko:

I'm pleased to inform you that your manuscript has been deemed suitable for publication in PLOS ONE. Congratulations! Your manuscript is now with our production department. 

Kind regards, 

on behalf of

Prof. Oleg Y. Dmitriev 

Academic Editor

PLOS ONE